# Correlations between Forgetfulness and Social Participation: Community Diagnosing Indicators

**DOI:** 10.3390/ijerph16132426

**Published:** 2019-07-08

**Authors:** Seungwon Jeong, Yusuke Inoue, Katsunori Kondo, Kazushige Ide, Yasuhiro Miyaguni, Eisaku Okada, Tokunori Takeda, Toshiyuki Ojima

**Affiliations:** 1Department of Community Welfare, Niimi University, Niimi, Okayama 718-8585, Japan; 2Faculty of Health and Welfare Science, Okayama Prefectural University, Soja, Okayama 719-1197, Japan; 3Department of Public Health, Graduate School of Medicine, Chiba University, Chiba, Chiba 260-8670, Japan; 4Department of Gerontological Evaluation, National Center for Geriatrics and Gerontology, Obu, Aichi 474-8511, Japan; 5Department of Rehabilitation, Hasegawa Hospital, Yachimata, Chiba 289-1113, Japan; 6Research Department, Institute for Health Economics and Policy, Minato-ku, Tokyo 105-0003, Japan; 7Department of Community Health and Preventive Medicine, Hamamatsu University School of Medicine, Hamamatsu, Shizuoka 431-3125, Japan; 8Department of Rehabilitation and Care, Seijoh University, Tokai, Aichi 476-0014, Japan

**Keywords:** social participation, social environment, community diagnosing, forgetfulness

## Abstract

We analyzed the relationships between forgetfulness and social participation, social contact, and social support by municipality to develop community diagnosing indicators. The analysis subjects included 105 municipalities that agreed to provide data for the 2013 Survey of Needs in Spheres of Daily Life in Japan (*n* = 338,659 people). Forgetfulness as a risk factor for dementia was used as the dependent variable. The variables of social environment factors were (1) social participation, (2) social contact, and (3) social support. The ratio of people responding that they experienced forgetfulness differed among municipalities, with a mean of 19.0% (7.1–35.6%). Higher levels of social participation, social contact, and social support were associated with lower levels of forgetfulness, even after adjusting for age and regional variables. The results of the present study suggest that it is appropriate to use forgetfulness and social participation at least a few times a year in any social activity as community diagnosing indicators. Municipalities could encourage their inhabitants to participate by developing and providing engaging social activities.

## 1. Introduction

According to the World Health Organization (WHO), social factors, such as social support, social networks, and social environment, are some determinants of human health. The Declaration of Alma-Ata (1978) states that health inequalities are unacceptable, and the WHO committee on the social determinants of health recommended improving the living environment in its final report in 2008. To promote social environment improvement in local regions, the WHO introduced an Urban Health Equity Assessment and Response Tool (Urban-HEART) [1] and age-friendly city indicators [2] as tools for evaluating the social environment and health inequalities in cities. In Japan, the Ministry of Health, Labour and Welfare (MHLW) proposed reducing health inequalities by developing social environments as part of its Health Japan 21 (the second term) program in 2013 [3]. However, which aspects of the social environment should be developed to reduce health inequalities, or which indicators should be used from among the community diagnosing indicators that measure the social environment remain unclear. 

Japan is aging at a faster rate than any other country, and its interest in the well-being of older people is growing. In Japan, there were over 4.62 million elderly with dementia in 2012, and the number is expected to increase to about 7 million in 2025, thus increasing the need for effective measures to prevent dementia [4]. However, the social-environmental factors that impact the prevention of dementia in an individual or in the community are unknown.

In this study, we validated the relevance of the indicator of forgetfulness as a regional diagnostic index. The correlation between forgetfulness (which was verified by the MHLW in Japan as a risk factor of the onset of dementia) and social participation, social contact, and social support was analyzed. 

## 2. Literature Review

### 2.1. Forgetfulness as a Risk of Dementia

Longitudinal studies of factors indicating the risk for dementia have identified forgetfulness as one factor. Many of these studies have focused particularly on biological factors or cognitive functions [5,6,7]. For example, several studies have shown that recognition of forgetfulness in oneself precedes a decline in cognitive function, indicating it as a risk of dementia [8,9]. Similarly, mild forgetfulness is observed in the early stages of dementia and may lead to reduced motivation and less interest in one’s surroundings [5,6].

### 2.2. Social Factors and Dementia

In general, social participation is considered to positively impact an individual’s health [10,11]. Studies on the prevention of the need for long-term care (LTC) have revealed inter-municipality differences in the ratio of social participation and twice the risk of functional disability (LTC need) [12,13], a lower rate of falls in regions with high ratios of participation in sporting groups [14], and a low rate of falls [15] and functional disabilities among older adults participating in sporting groups or other such groups at the municipality [16] and individual levels [17]. For dementia, studies have been conducted to examine whether the associations with social relationships [18,19,20,21,22,23,24], social contact [21,22], social support [10,11], and other forms of social participation, as psychosocial factors, are linked to the onset of dementia [25,26].

### 2.3. Community Diagnosis

Community diagnosis refers to issues and priorities for each region through the observation of target regions (municipalities, prefectures, etc.) and health care statistics to develop evidence-based health policy and public health at the regional level to clarify the countermeasure focus areas [27].

The WHO Urban Health Equity Assessment and Response Tool (Urban HEART) and the Community-Based Integrated Care: Visualization System [28] developed by MHLW are some of the types of community diagnosing tools.

The percentage of Japanese aged 65 and older was 28.1% in 2018. Therefore, it is necessary to develop effective care prevention measures at the regional level. To do so, it was first necessary to have a good understanding of each region; thus, community diagnosis was implemented across the municipalities. 

The community diagnostic indicators implemented mainly use the LTC risk index related to care prevention and the index related to the certification of needed LTC. Although there have been many studies on depression [29,30,31,32], oral function problem [33,34,35,36], and being housebound (Tojikomori) [37] among the six risks of LTC indicated by the MHLW (motor function decline, malnutrition, oral function problem, Tojikomori (housebound), declining accreditation function, depression), few studies have been conducted on community diagnosing indicators to measure dementia risk or health inequalities in dementia risk.

## 3. Study Methods

### 3.1. Subjects

The analysis subjects included 109 municipalities that agreed to provide data for 2013 for the JAGES (Japan Gerontological Evaluation Study) HEART [38] program or the Survey of Needs in Spheres of Daily Life for the elderly (The Survey of Needs) [39]. The Survey of Needs is a survey in which the MHLW presented models that were then implemented by numerous municipalities to obtain basic information for creating plans for LTC insurance programs. The Survey of Needs examined a random sample of adults aged 65 or older who had not been certified for LTC for each sub-region for the spheres of daily life set for each municipality once every three years. In the 2013 survey, 246 municipalities (cities, towns, villages) of about 1700 municipalities nationwide participated. In this study, 109 municipalities cooperated among the 246 municipalities (44.3% of the surveyed subjects). The survey was distributed and collected by mail. Data were requested from municipalities through a website [40] launched by the MHLW Scientific Research Team and the sixth meeting of personnel involved in the formulation and preparation of plans for LTC insurance programs conducted by the Health and Welfare Bureau for the Elderly, MHLW. 

Excluding four municipalities that were outliers by more than two standard deviations, the data were sourced from 357,850 people in 109 municipalities. The final sample included 338,659 older adults—152,823 men, and 185,163 women (missing 673)—with ages ranging from 65 to 106 (mean 74.35, SD 6.65) from 105 municipalities. As we did not obtain information on the survey response rates for each municipality and subregion, we used data on a municipality basis without weighting.

### 3.2. Variables Used in the Analysis

Forgetfulness is considered a risk factor for dementia onset [41]. Those who responded “Yes” to the question posed by MHLW [39], “Do people tell you that you are forgetful?” were labeled as those who were deemed forgetful. Variables of social environment factors were: (1) social participation (participation in volunteer groups, sports groups, interest groups, senior citizens’ clubs, neighborhood or residents’ associations, study or cultural groups, activities to support older adults requiring protection, activities to support parents raising children, and community living environment improvement (beautification) activities, or paid work); (2) social contact (contact with friends and/or acquaintances); and (3) social support (providing or receiving emotional support, providing or receiving instrumental support, receiving emotional or instrumental support, providing emotional or instrumental support).

Respondents were asked for the frequency of social participation and social contact items. Each item was categorized into: participates more than once a week, participates more than once a month, participates more than a few times a year, and does not participate. In social support, we measured whether they did or did not receive support.

To adjust for regional factors, we used (1) the ratio of older adults living alone, (2) the population density of the inhabitable land area, (3) the ratio of older adults educated under nine years, and (4) the taxable income as control variables that may affect social environment variables [16], as shown in Table 1.

### 3.3. Analytical Methods

#### 3.3.1. Age Adjustment

A simple comparison of the rate of forgetfulness and rate of social participation by region shows differences in the age composition of each region. As a result, regions with a larger older adult population tended to have poorer health outcomes and, conversely, regions with fewer older adults tended to have better health outcome results. To enable a comparison of the various indicators among regions with different age compositions, we applied the age standardization method used by the MHLW to adjust the age for all variables in this study. Using these age-adjusted variables, we were able to compare regions more accurately without paying attention to the age composition in groups with different age compositions. The calculation equation is as follows [42]: Regional age adjustment index rate = {(Regional Coarse index^※^ rate by age of 5 years ×The population of the age group of reference population) ×The sum total of each age class of the indicator} / Total number of reference population(1)

^※^ The coarse index rate obtained by dividing each indicator by the population.

#### 3.3.2. Statistical Method

We examined the extent of inter-municipality differences in the ratio of people with forgetfulness and in social environment factors (social participation, social contact, and social support). We then calculated the partial correlation coefficients for the relationships between the ratio of people with forgetfulness and (1) social participation, (2) social contact, and (3) social support, adjusting for the ratio of older adults living alone, the population density of the inhabitable land area, the ratio of older adults educated for less than nine years, and the taxable income as control variables. SPSS Statistics for Windows, Ver. 24 (IBM, Armonk, NY, USA) was used for the analysis.

### 3.4. Ethical Considerations

The study protocol was approved by the Ethics Committees of Nihon Fukushi University (13-14), Chiba University (1777, 2493) and the National Center for Geriatrics and Gerontology (No. 992, No. 1028-2).

## 4. Results

### 4.1. Inter-Municipality Differences

#### 4.1.1. Ratio of People with Forgetfulness

In the total data of the 338,659 people in the 105 municipalities, 22,734 (12.8%) responded saying that they were seen by others as forgetful. The mean ratio of people with forgetfulness by municipality was 19.0% (7.1–35.6%), a significant 5.1-fold difference between the municipalities with the lowest and highest ratios. 

#### 4.1.2. Ratio of Social Participation, Social Contact, and Social Support

The ratio of social participation of a few times a year or more, after adjusting for age, was highest for (1) neighborhood or residents’ associations at 45.3% (14.3–75.1%), followed by (2) interest groups at 38.5% (24.2–49.2%), (3) community living environment improvement (beautification) activities at 33.7% (9.8–59.8%), (4) sports groups at 28.8% (14.5–42.4%), (5) paid work at 27.4% (13.9–43.1%), and (6) volunteer groups at 25.3% (7.7–45.5%). The ratio of social participation of “participate in at least any group” for “a few times a year or more” was 75% (47.3–89.1%). 

As an indicator of social contact, the frequency of networking with friends and/or acquaintances a few times a year or more was 90.5% (72.5–96.4%). 

With regard to social support, 96.7% (90.6–98.8%) had someone they could talk to about worries or grievances (providing or receiving emotional support), and 96.1% (90.0–98.0%) had someone who would care for them if they fell ill or look after them (providing or receiving instrumental support). The ratio of those receiving some type of support was 98.2% (90.6–99.4%), and the ratio of those providing some type of support was 95.0% (88.5–97.5%) (Table 1).

### 4.2. Partial Correlation Analysis between the Ratio of People with Forgetfulness and Social Participation by Municipality

From the data at the municipality level, we calculated the partial correlation coefficients (*r*) for the correlations between the ratio of people with forgetfulness and social participation, social contacts, and social support, and adjusted the ratio of older adults living alone, the population density of the inhabitable land area, the ratio of older adults educated for less than nine years, and taxable income by municipality. 

The ratio of people with forgetfulness was lower in areas with more social participation, social contact, and social support.

Significant negative correlations were observed between the ratio of people with forgetfulness and the ratio of people who socially participated: a few times a year or more in neighborhood or residents’ association (*r* = −0.55), community living environment improvement activities (*r* = −0.54), interest groups (*r* = −0.51), sports groups (*r* = −0.48), study or cultural groups (*r* = −0.46), and volunteer groups (*r* = −0.43). In addition, significant negative correlations were observed between the ratio of people with forgetfulness and the ratio of people who socially participated in one or more of the items showed statistically significant results (*r* = −0.72) (Figure 1).

With regard to social contact, a negative correlation was observed with a few times a year or more (*r* = −0.66).

For social support, negative correlations were observed with providing or receiving emotional support (*r* = −0.45) and providing or receiving instrumental support (*r* = −0.38) (Table 2).

## 5. Discussion

### 5.1. High Ratio of Social Participation in Regions with a Small Ratio of People with Forgetfulness

In this study, we analyzed the associations between forgetfulness and social participation, social contact, and social support by municipality using data from 105 municipalities that participated in the Survey of Needs conducted with adults aged 65 and older who were not certified for LTC.

The ratio of people who experienced forgetfulness differed among municipalities by a maximum of 28.5% points (ranging from 7.1 to 35.6%), which is a 5.1-fold difference between the municipalities with the highest and lowest ratios. Higher levels of social participation, social contact, and social support were associated with lower levels of forgetfulness, even after adjusting for age and regional variables. The difference of up to 5.1 times among the 105 municipalities, even after adjusting for age and regional variables, indicates the need to measure health inequalities at the regional level and that forgetfulness may be related to factors regarding the social environment at the regional level. 

Some previous individual-level analytical studies on the risk of dementia revealed that social relations [18,19,20,21,22], productive social engagement [43], and leisure activities [44] are associated with cognitive decline or the onset of dementia. We found a negative correlation between the ratio of people with forgetfulness and the ratio of social participation at the regional level. Thus, social environment (social participation, social contact, social support) was correlated with forgetfulness that further develops into early dementia, not only for individuals, but also communities. Through this result, we can suggest that forgetfulness is an appropriate indicator for community diagnosis.

### 5.2. Appropriateness as a Community Diagnosing Indicator

Among the indicators we tested, the social participation indicators had a higher correlation coefficient with forgetfulness than social contact or social support. 

For social participation, we analyzed what kind of items are correlated with forgetfulness and how often social participation correlated to forgetfulness at a regional level. As a result, although the magnitudes of the correlation coefficients differed, analysis by the type of social participation revealed negative correlations with forgetfulness for participation in neighborhood or residents’ associations, community living environment improvement (beautification) activities, interest groups, sports groups, study or cultural groups, volunteer groups. This finding is consistent with those of previous studies conducted in different regions. For example, negative correlations were observed at the level of insured individuals (municipalities and medical care associations) [32] and elementary school zones [22] between several functional disability risk factors [45] and the ratio of participants in sports groups [33]. Negative correlations were also observed between the ratio of older adults participating a few times a year in sports or interest groups and the LTC eligibility rate [21] and the suicide rate of older adults [46]. Significant negative correlations were observed between the ratio of people with forgetfulness and social participation in any social activity. This means that there were fewer people who experienced forgetfulness in regions where a large ratio of people participated more in any kind of social activity. Regardless of the type and frequency of social participation, participation in any kind of social activity may lead to the prevention of forgetfulness.

In terms of frequency, we found a stronger correlation with forgetfulness for social participation “a few times per year or more” than “once a week or more” or “once a month or more.” This suggests that social participation a few times a year or more is good enough as a community diagnosing indicator of social participation frequency. No studies to date have clarified the difference in the magnitudes of the correlation coefficients depending on the social participation frequency at the regional level. In the present study, the social participation frequency of a few times a year or more was significantly associated with most variables, and the correlations coefficients were high, suggesting that the ratio of older adults participating in some type of activity a few times a year or more is appropriate as a community diagnosing indicator. On an individual level, it is likely that a higher frequency of participation has greater effects; therefore, different indicators should be used for individual and regional level analyses. This implies that there is a need for research to be performed at the regional level, independent of individual-level analyses. Thus, for a community diagnosis, municipalities could encourage its people to participate in social activities.

### 5.3. Significance and Limitations of the Study and Future Research

Up to this point, many studies have focused on social participation related to health determinants. However, no regional level study has been conducted for the prevention of cognitive decline, although the need for such a study in Japan has been much emphasized. It was not easy to obtain data for these studies, and the index for community diagnosis was insufficient. In the present study, we analyzed the relationship between forgetfulness and social environment using the existing regional data to provide a regional diagnosis. The results of our study suggest that it may be possible to provide insights for regional intervention aiming to create an age-friendly region when searching for social factors with a low ratio of people with forgetfulness. 

This study also has its limitations. The data we used were not originally designed for this study, and were previously collected from municipalities; therefore, we could not control the sampling. The questionnaires and definitions require more discussion. We controlled the data using four factors of regional variables (ratio of older adults living alone, population density of the inhabitable land area, ratio of older adults educated for less than nine years, and taxable income by municipality) related to social activities for older people, but other factors could be related that we did not include in this study. 

Further studies may be needed to verify the appropriateness of the community diagnosing indicator at the sub-region level by analyzing the spheres of daily life and subregions. In addition, to develop a community diagnosing indicator that encompasses more diversified perspectives, further studies are required to test the relationships between forgetfulness and other social factors that were not considered in the present study. Longitudinal studies could be conducted to determine causal relationships and examine temporal changes on a regional level. 

## 6. Conclusions

In the present study, we used data collected through the Survey of Needs from 105 municipalities to analyze the relationships between forgetfulness and social participation and other factors by municipality to develop community indicators. 

The results revealed a difference of more than 5.1 times among the municipalities in the ratio of people with forgetfulness (7.1–35.6%), and differed almost two times (47.3–89.1%) in terms of the social participation in any activity a few times a year or more among the regions. The correlation between the ratio of forgetfulness and the ratio of social participation showed a negative correlation, which means that those who participate socially are less likely to develop forgetfulness. We suggest that it is appropriate to use forgetfulness and social participation at least a few times a year in any social activity as community diagnosing indicators that can be used for dementia prevention plans. Monitoring the numerical value of these indicators, even at regional levels smaller than the municipality, would also be effective.

## Figures and Tables

**Figure 1 ijerph-16-02426-f001:**
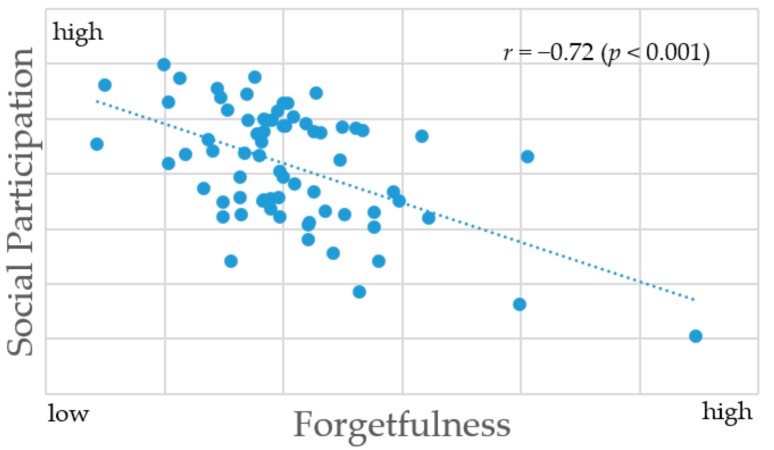
Scatter plot of the partial correlation between forgetfulness and social participation by municipality. Forgetfulness: the ratio of people with forgetfulness by municipalities. Social Participation: the ratio of people taking part in any activities a few times a year or more by municipality.

**Table 1 ijerph-16-02426-t001:** Variables used in the analysis and means and ranges of ratios by municipality.

Variables		Age-Adjusted (*n* = 105 Municipalities)		
Frequency		A Few Times a Year or More% (Range)	Once per Month or More% (Range)	Once per Week or More% (Range)
**Forgetfulness**	Those who are deemed forgetful	19.0 (7.1–35.6)		
**Social Participation**	Neighborhood or residents’ association	45.3 (14.3–75.1)	2.5 (0.5–7.1)	2.4 (0.5–5.4)
	Interest club or groups	38.5 (24.2–49.2)	28.9 (12.4–40.8)	21.5 (9.8–30.7)
	Community living environment improvement activities	33.7 (9.8–59.8)	4.9 (1.5–11.9)	1.6 (0.5–4.3)
	Sports club or groups	28.8 (14.5–42.4)	22.1 (6.2-34.0)	16.6 (1.9-26.8)
	Paid work	27.4 (13.9–43.1)	22.6 (12.3–35.5)	19.5 (11.4–31.9)
	Volunteer club or groups	25.3 (7.7–45.5)	13.2 (3.5–21.0)	6.1 (1.8–10.4)
	Senior citizens’ club	24.0 (6.0–49.6)	9.9 (3.7–19.2)	3.5 (0.6–10.3)
	Study or cultural groups	16.9 (8.5–30.1)	9.0 (1.8–26.2)	3.6 (0.313.8)
	Support for older adults requiring protection	10.1 (4.5–31.7)	5.3 (2.2–14.6)	2.8 (1.0–9.5)
	Support for older adults requiring nursing care	7.2 (3.9–19.9)	3.2 (1.2–6.8)	2.1 (0.9–4.5)
	Support for parents raising children	7.5 (3.8–13.1)	3.7 (1.2–8.0)	2.3 (0.4–6.1)
	Participate in any group above	75.0 (47.3–89.1)	55.2 (32.7–69.6)	41.6 (22.3–54.3)
**Social Contact**	Contact with friends and/or acquaintances	90.5 (72.5–96.4)	72.1 (52.2–86.3)	49.5 (30.8–69.5)
**Social Support**	Providing or receiving emotional support	96.7 (90.6–98.8)		
	Providing or receiving instrumental support	96.1 (90.0–98.0)		
	Receiving emotional or instrumental support	98.2 (90.6–99.4)		
	Providing emotional or instrumental support	95.0 (88.5–97.5)		
**Regional Variables**	Ratio of older adults living alone ^1^	10.1% (5.0%–28.3%)		
	Population density of the inhabitable land area ^1^	1749.3 people (3.4–18,253.7 people)		
	The ratio of older adults educated for less than 9 years ^1^	46.6% (11.4%–84.0%)		
	Taxable income ^2^ (Yen)	290,075.1 (536.0–6,817,509)		

^1^ Estimates from the 2010 National Census. Calculated by dividing by the number of older adults living alone, working, or less than nine years of education. ^2^ Estimated based on municipality circumstances observed in the statistics (2010). Calculated by dividing the taxable income by the number of taxpayers (per-income levy).

**Table 2 ijerph-16-02426-t002:** Partial correlation analysis between the ratio of people with forgetfulness and social participation by municipality.

Variables		Correlation Coefficient (*r*)		
Frequency		A few Times a Year or More	Once per Month or More	Once per Week or More
		Experiences Forgetfulness		
**Social Participation**	Neighborhood or residents’ association	−0.55 **	−0.21	−0.20
	Interest groups	−0.51 **	−0.38 *	−0.37 *
	Community living environment improvement activities	−0.54 **	−0.20	−0.04
	Sports groups	−0.48 **	−0.39 **	−0.26
	Paid work	−0.25	−0.21	−0.15
	Volunteer groups	−0.43 **	−0.40 **	−0.31
	Senior citizens’ club	−0.27	−0.14	0.10
	Study or cultural groups	−0.46 **	−0.29	−0.29
	Support for older adults requiring protection	−0.22	−0.04	−0.07
	Support for older adults requiring nursing care	−0.23	−0.11	−0.19
	Support for parents raising children	−0.24	−0.20	−0.13
	Participate in any group above	−0.72 **	−0.68 **	−0.64 **
**Social Contact**	Contact with friends and/or acquaintances	−0.66 **	−0.36 **	−0.17
**Social Support**	Providing or receiving emotional support	−0.45 **		
	Providing or receiving instrumental support	−0.38 *		
	Receiving emotional or instrumental support	−0.23		
	Providing emotional or instrumental support	−0.13		

Note: Adjusted for the ratio of older adults living alone, the population density of the inhabitable land area, the ratio of older adults educated for less than nine years, and taxable income by municipality. * *p* < 0.05, ** *p* < 0.001.

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
