# Peer review of "Correlations between Forgetfulness and Social Participation: Community Diagnosing Indicators"

_ijerph, 2019, doi:10.3390/ijerph16132426_

Round 1

Reviewer 1 Report

This is an important topic and clearly much needs to be done to reduce dementia in older adults in all cultures.

I recommend a section titled Literature Review to come after the introduction. Anchoring this work in the literature will be helpful.

The article is written at a very high level that is not readily accessible to most readers and really seems aimed only at other researchers. As you revise your work, think about how to make it appeal to clinicians, policy makers, and others who can translate research results into improved policies and programs.

Author Response

Thank you very much.

We appreciate your comments and helpful suggestions, which have helped us to considerably improve our paper.

We added Literature Review and included “diagnosing indicator” and related factors.

Also, we got the Professional language editing services by MCPI.

Please see the revision.

Reviewer 2 Report

Minor comments:

Change the following sentence in the abstract: "The ratio of people with forgetfulness correlated negatively with higher ratios of social participation, social contact, and social support after adjusting for age and regional variables." 

Do you mean: the ratio of people with XX is negatively correlated with ratios of social participation..... (that is, higher level of social participation is associated with lower levels of forgetfulness?)

Please get a professional editor to edit the manuscript. Some awkward sentences and grammar mistakes throughout - e.g :

" A previous individual-level analysis study on risk of dementia revealed that social relations are associated with forgetfulness or the onset of dementia" ( but the references refer to multiple studies);

"Thus, social environment (social participation, social contact, social support) is correlated with forgetfulness further develop to early dementia not only individual-level but also region-level" (??);

"This suggests that any kind of social activities can be associated in forgetfulness " (??);

"..we revealed more stronger correlation with forgetfulness for those participating" (grammar). 

        A thorough English editing will significantly improve the clarity of the manuscript. 

More substantive comments: 

" Currently in Japan, community diagnosis has been recommended to be conducted but it takes much time and expense.". What do you mean by community diagnosis? Please expand for readers who are not familiar with this. 

The paper will be much strengthened with a graphical representation of the variation in forgetfulness/social participation (e.g. representing the ratio across municipalities/major regions through a map). That way readers can identify whether there are some geographical clustering around these variables

Most of the references cite papers based on Japanese data ( as it should be) and other studies based on Western contexts. How does the study's key  result compare to studies on social participation and dementia in Japan's neighbouring countries and immediate region?  (East Asia and Southeast Asia)?  See for example: Lee, Y., & Yeung, W. J. J. (2019). Gender matters: Productive social engagement and the subsequent cognitive changes among older adults. Social Science & Medicine229, 87-95; Leung, G. T. Y., Fung, A. W. T., Tam, C. W. C., Lui, V. W. C., Chiu, H. F. K., Chan, W. M., & Lam, L. C. W. (2011). Examining the association between late‐life leisure activity participation and global cognitive decline in community‐dwelling elderly Chinese in Hong Kong. International journal of geriatric psychiatry26(1), 39-47)

Readers would also be interested to know what really drives the regional/inter-municipalities variation in social participation. The authors already standardised for age structure, but  there would be other regional level variables that would affect levels of social participation. If this is beyond the scope of the paper, at least include this in the limitation section, noting findings from recent studies based on Asian data (see Utomo, A., Mcdonald, P., Utomo, I., Cahyadi, N., & Sparrow, R. (2019). Social engagement and the elderly in rural Indonesia. Social Science & Medicine229, 22-31.)

Best wishes to the authors. Thanks. 

Author Response

Thank you very much.

We appreciate your comments and helpful suggestions, which have helped us to considerably improve our paper.

We got the Professional language editing services by MCPI.
